# SARS-CoV-2 Variant of Concern 202012/01 Has about Twofold Replicative Advantage and Acquires Concerning Mutations

**DOI:** 10.3390/v13030392

**Published:** 2021-03-01

**Authors:** Frederic Grabowski, Grzegorz Preibisch, Stanisław Giziński, Marek Kochańczyk, Tomasz Lipniacki

**Affiliations:** 1Institute of Fundamental Technological Research, Polish Academy of Sciences, 02-106 Warsaw, Poland; frederic.grabowski@ippt.pan.pl; 2Inter-Faculty Individual Studies in Mathematics and Natural Sciences, The MISMaP College, University of Warsaw, 02-097 Warsaw, Poland; grzegorz.preibisch@gmail.com; 3Faculty of Mathematics, Informatics and Mechanics, University of Warsaw, 02-097 Warsaw, Poland; sg385513@students.mimuw.edu.pl

**Keywords:** COVID-19 pandemic, SARS-CoV-2, spike protein, VOC-202012/01, spike L18F, genome sequencing, mutation

## Abstract

The novel SARS-CoV-2 Variant of Concern (VOC)-202012/01 (also known as B.1.1.7), first collected in United Kingdom on 20 September 2020, is a rapidly growing lineage that in January 2021 constituted 86% of all SARS-CoV-2 genomes sequenced in England. The VOC has been detected in 40 out of 46 countries that reported at least 50 genomes in January 2021. We have estimated that the replicative advantage of the VOC is in the range 1.83–2.18 [95% CI: 1.71–2.40] with respect to the 20A.EU1 variant that dominated in England in November 2020, and in range 1.65–1.72 [95% CI: 1.46–2.04] in Wales, Scotland, Denmark, and USA. As the VOC strain will likely spread globally towards fixation, it is important to monitor its molecular evolution. We have estimated growth rates of expanding mutations acquired by the VOC lineage to find that the L18F substitution in spike has initiated a fast growing VOC substrain. The L18F substitution is of significance because it has been found to compromise binding of neutralizing antibodies. Of concern are immune escape mutations acquired by the VOC: E484K, F490S, S494P (in the receptor binding motif of spike) and Q677H, Q675H (in the proximity of the polybasic cleavage site at the S1/S2 boundary). These mutants may hinder efficiency of existing vaccines and expand in response to the increasing after-infection or vaccine-induced seroprevalence.

## 1. Introduction

The earliest genome belonging to the novel SARS-CoV-2 Variant of Concern (VOC)-202012/01, also known as B.1.1.7 lineage, was collected on 20 September 2020, in Kent, UK (GISAID sequence accession ID: EPI_ISL_601443). The lineage, characterized by nine spike protein mutations (deletions: 69–70 HV, 145V; substitutions: N501Y, A570D, D614G, P681H, T716I, S982A, D1118H), started to spread rapidly in mid-October 2020 to constitute in January 2021 86% of all SARS-CoV-2 genomes sequenced in England [1,2]. Spread of the VOC-202012/01 variant, hereafter referred to as the VOC, co-occurred with a rapid surge of cases in December in Kent and Greater London [3,4].

In England, the VOC is currently replacing the recently dominant 20A.EU1 strain, characterized by A222V substitution in spike protein [5]. Both strains are independent substrains of a spike glycoprotein D614G variant that has spread in spring 2020 in England and worldwide, almost reaching fixation [6]. The 20A.EU1 strain started expanding in England in mid-August 2020 and constituted more than 65% of genomes sequenced in England in November 2020 [2]. The VOC strain in the receptor-binding motif (RBM) of spike shares mutation N501Y with the 501Y.V2 and P.1 strains that are currently rapidly spreading in South Africa and Brazil, respectively [7,8].

Deletion 69H–70V in spike glycoprotein, which is characteristic, but not unique, to the VOC, prevents detection of the spike gene by the dPCR probe used by some laboratories of the English diagnostic system (spike gene target failure, SGTF) [9]. As the VOC has a high multiplicative potential, it has become the most prevalent Δ69–70HV strain; consequently, the proportion of the SGTF has been used as a proxy for the prevalence of the VOC genome [9]. Based on the SGTF, the Public Health England agency determined the multiplicative advantage of the VOC in NHS STP areas of England in weeks 44–48 of 2020. As an average for the considered STP areas the authors obtained the ratio of reproduction numbers equal 1.47 [95% CI: 1.34–1.59] [9]. Leung et al., based on GISAID data from the period 22 September –16 November 2020, and a competition transmission model of two viruses, estimated this ratio as 1.75 [95% CrI: 1.70–1.80] [10]. Davies et al., based on data from the COVID-19 Genomics UK (COG-UK) Consortium from October and November 2020 estimated that the VOC is 43–82% [95% CrI: 38–106%] more transmissible than preexisting variants of SARS-CoV-2 [11,12]. Our early estimate of the ratio of reproduction numbers of the VOC to non-VOC strains, based on GISAID data in weeks 43–47 of 2020 in England was 2.24 [95% CI: 2.03–2.48] [13].

In this study, based on GISAID data available on 12 February 2021 (when this manuscript has been revised), we calculated the growth of the ratio of the VOC to the 20A.EU1 genome sequences collected in the period between week 43 and week 51 of 2020 (19 October–20 December 2020) and estimated the replicative advantage of the VOC strain in relation to the 20A.EU1 strain. Then, using the same approach, we estimated the replicative advantage of the 20A.EU1 strain in relation to previous D614G strains and the D614G strain in relation to the D614 strains (i.e., the strains with non-mutated residue 614). In such approach, we analyze the progression of strains with increasing replication advantage: D614G → 20A.EU1 → the VOC. In contrast to the approach of Public Health England (PHE) agency [9] and Davies et al. [11], we used England-aggregated weekly data. We prefer to use such aggregated data because of the geographically non-uniform emergence of VOC substrains that, as we will show, have diverse replication advantages. Some substrains that emerge locally may have low or no replicative advantage and in the long term may be out-competed and eventually eliminated; however, such transient substrains can considerably contribute to the genome composition in specific NHS STP areas of England.

The VOC strain, as all strains, mutates continuously. Because of its significant replicative advantage, any accrued mutations gain an opportunity to spread (potentially worldwide), depending on their replicative advantage with respect to the bulk VOC strain, or higher ability to infect seroprevalent individuals. We have systematically estimated growth rates of spreading mutations acquired by the VOC lineage to find that spike L18F substitution has likely initiated a substrain of replicative advantage. The L18F mutation is of significance because when recently analyzed in the context of the South African strain 501Y.V2 it has been found to compromise binding of neutralizing antibodies [14,15].

## 2. Materials and Methods

### 2.1. Data Sources

In Figure 1, Figure 2, Figure 3, Figure 4, Figure 5, Figure 6 and Figure 7 we use weekly aggregated data of samples collected in England or South Africa submitted to GISAID until 12 February 2021, and samples collected in Wales, Scotland, Denmark, and the USA and submitted until 17 February 2021. Table 1 is based on GISAID data as of 16 February 2021. In Appendix A, we use COG-UK pillar 2 data as of 15 February 2021. All data used for Figure 1, Figure 2, Figure 3, Figure 4, Figure 5, Figure 6 and Figure 7 and Appendix A are collected in a single file, Appendix A.

### 2.2. Genome Sequence Analysis

We analyzed mutations in spike gene sequences that had at most 5% letters other than A, C, G, T. Sequences devoid of full daily collection date were excluded from the analysis. The spike gene was localized using EMBOSS stretcher and then re-aligned using EMBOSS needle [16] to the reference sequence: GenBank’s NC_045512.2 from Wuhan or GISAID’s EPI_ISL_601443 from Kent. To minimize ambiguous reporting of mutations variants, indels were left-aligned using an in-house script; insertions and deletions on consecutive residues were collated and considered a single mutation.

### 2.3. Monte Carlo Estimation of the Confidence Interval for the Growth
Rate of the Ratio of Strains

To estimate the 95% credible interval for *k* subsequent weeks *a priori* we assume that genomes of two compared strains in each week of the considered period follow a respective binomial distribution having a success probability *p* = $n_{x}$/($n_{x}$ + $n_{y}$), where $n_{x}$ and $n_{y}$ are the numbers of two compared strain genomes (e.g., VOC and 20A.EU1). By sampling from *k* such binomial distributions for the considered time window of *k* weeks $10^{5}$ times, we obtained $10^{5}$ series of *k* simulated sequenced genome proportions. We performed fitting to each such series to obtain $10^{5}$ estimates of the weekly growth rate of the ratio of the genome sequences. In all cases but one for L18F analysis (Figure 7) the 95% credible interval obtained using the *a priori* method has been narrower than the 95% confidence interval calculated as 1.96× (standard error of the slope). In these cases we reported confidence interval, while in L18F case we reported the credible interval.

### 2.4. Estimation of the Relative Replication Advantage of Viral Strains

To estimate the ratios of replication numbers of strain *x* and *y* we estimated the weekly growth of the ratio of the number of their sequenced genomes, *v*. Then, assuming that both strains have the same average serial interval of 6.73 days [17], we obtained the ratio of their replication numbers $R_{t}^{x}/R_{t}^{y}$
:=$v^{6.73/7}$. This estimation of the relative replicative advantage of viral strains does not involve a direct calculation of their reproduction numbers.

## 3. Results

### 3.1. Evolution of SARS-CoV-2 in England

The first prevailing mutation of SARS-CoV-2 was D614G substitution in spike protein (the first GISAID reported genome, EPI_ISL_913915, was collected on 2 January 2020, in Mexico). This substitution initiated a strain that spread worldwide nearly reaching global fixation [6]. In England, the D614G strain appeared during the spring wave of epidemic in 2020, and in summer it exceeded 98% of all sequenced genomes (Figure 1). Its substrain, 20A.EU1, started expanding in England in week 31 of 2020 and reached its maximum of 68% of all sequenced genomes in week 44 (Figure 1). The VOC strain is also a substrain of the D614G lineage, independent of 20A.EU1, which started expanding in week 43 (before week 43, less than 5 VOC genomes were collected per week), and in week 51 reached 57% of all sequenced genomes. At the same time the proportion of the 20A.EU1 strain dropped to 35% and the proportion of all other genomes dropped to 8%. One can observe that in weeks 44–51 the decrease of the proportion of 20A.EU1 is about twofold (from 68% to 35%), while the decrease of the proportion of remaining strains (that is, strains other than considered 20A.EU1 and VOC) is nearly fourfold (from 31% to 8%). This motivates us to calculate the replicative advantage of the VOC strain in relation to the 20A.EU1 strain instead of all non-VOC strains (the replicative advantage of the VOC over non-VOC strains would depend on the proportion of the 20A.EU1 strain in all non-VOC genomes).

### 3.2. Replicative Advantage of the VOC Strain over the 20A.EU1 Strain in
England

In the eight-week period of weeks 43–51, the ratio *q* of the VOC to the 20A.EU1 strain increased from *q* = 30/4030 ≈ 0.0074 to *q* = 4928/3051 ≈ 1.62, that is, 217 times. This implies that *q* increased $217^{1/8}$≈ 1.96-fold per week. To estimate the growth of *q* in a more rigorous way, we fitted the trend line using two fitting windows (Figure 2A). Fitting in weeks 43–47 gives weekly growth *v* = 2.24 [95% CI: 2.03–2.48], whereas fitting in weeks 43–51 gives *v* = 1.88 [95% CI: 1.75–2.01]. These confidence intervals are estimated as 1.96 × standard error of the slope. However, as the estimate for weeks 43–47 is based on only 5 data points and the number of the VOC genomes in the first data point is small (*n* = 30), we additionally estimated the 95% credible interval for weeks 43–47 *a priori* (assuming binomial distributions of VOC and 20A.EU1 strain genomes, see Methods). Using this auxiliary method, we estimated that the 95% CrI is 2.09–2.45, which is somewhat narrower than the 95% CI calculated from the standard error of the slope. This demonstrates that the CI calculated from the standard error of the slope was not a result of an incidental linearity of data points.

To estimate the ratio of reproduction numbers of the VOC and the 20A.EU1 strain, $R_{t}^{\mathrm{VOC}}/R_{t}^{20A.\mathrm{EU}1}$, we assumed that both strains have the same average serial interval of 6.73 days [17]. Then, $R_{t}^{\mathrm{VOC}}/R_{t}^{20A.\mathrm{EU}1}$ = $2.24^{6.73/7}$ = 2.18 [95% CI: 1.97–2.40] for the fitting window in between weeks 43 and 47, and $R_{t}^{\mathrm{VOC}}/R_{t}^{20A.\mathrm{EU}1}$ = $1.88^{6.73/7}$ = 1.83 [95% CI: 1.71–1.96] for fitting window in between weeks 43 and 51. The eight-week window estimate gives a smaller advantage of the VOC than the four-week window. We think that this discrepancy is caused by two factors. As shown in the report of PHE [3] in week 51 of 2020, the VOC almost reached fixation in the Greater London and Kent, being nearly absent in central England. Additionally, more stringent measures implemented in Greater London and Kent limited the growth of absolute numbers of VOC cases. The heterogeneity in ratio *q* is not important as long as *q* is low (as it was until week 47); however, when *q* becomes high in some subregions, growth of *q* in the whole region decelerates. For this reason, we have not extended the fitting window past week 51. Since we may not rule out a chance that the higher growth of *q* in the window of 43–47 weeks is an artifact caused by a small number of data points, we conclude that that ratio $R_{t}^{\mathrm{VOC}}/R_{t}^{20A.\mathrm{EU}1}$ is in the range 1.83–2.18 [95% CI: 1.71–2.40].

We should notice that sequenced genomes are submitted to GISAID with some time delay after sample collection; however, as of 12 February, the data for weeks 44–51 of 2020 appears nearly complete. As shown in Figure 2B, replicative advantage estimates stabilize with the progression of the ‘date of last submission’. We have also verified our fits using the COG-UK genome database. We performed the analysis based on pillar 2 genomes that excludes routine tests of health and care workers and other tests made for particular purposes (pillars 1, 3, and 4) (https://www.gov.uk/government/publications/coronavirus-covid-19-testing-data-methodology/covid-19-testing-data-methodology-note, accessed on 12 February 2021). For the COG-UK data we obtained nearly the same (±3%) replicative advantage of the VOC strain in the 43–47 and 43–51 week windows, see Appendix A for fits and values.

In the same way we estimated that the ratio of $R_{t}^{}$ of the VOC strain to $R_{t}$ of other strains (that are neither VOC nor A20.EU1) is in the range of 2.03–2.47 [95% CI: 1.89–2.77], where the lower bound is the estimate within weeks 43–51 and the higher bound is the estimate for weeks 43–47 of 2020. As may be expected, $R_{t}^{\mathrm{VOC}}/R_{t}^{\mathrm{other}}$ > $R_{t}^{\mathrm{VOC}}/R_{t}^{20A.\mathrm{EU}1}$, which reflects the fact that the A20.EU1 strain has a replicative advantage over previously dominant non-VOC D614G strains. We estimated this advantage by fitting a trend line to data points from weeks 34–45 of 2020, because for this period the exponential growth of the ratio of the A20.EU1 strain to other non-VOC D614G strains is observed (Figure 2C). In this period, the ratio of the A20.EU1 to other non-VOC D614G strains grows at a rate of 1.25 [95% CI: 1.23–1.28] per week, which gives $R_{t}^{20A.\mathrm{EU}1}/R_{t}^{\mathrm{other}\;\;\mathrm{non}\text{-}\mathrm{VOC}\;\;D614G}$ = $1.25^{6.73/7}$ = 1.24 [95% CI: 1.22–1.27]. Finally, we estimated that the ratio $R_{t}^{D614G}/R_{t}^{D614}$= $1.44^{6.73/7}$ = 1.42 [95% CI: 1.38–1.45]. Concluding, we showed that the D614G strain that spread worldwide towards fixation had replicative advantage of 1.42 in relation to D614 strains in England. Its substrain A20.EU1 had replicative advantage of 1.24 over bulk D614G, and reached the proportion of 68% of genomes in England. Currently, A20.EU1 is outcompeted by the VOC that has about two-fold replicative advantage in relation to the A20.EU1 strain.

### 3.3. Worldwide Spread of the VOC Strain

London serves as a major transportation hub and thus, unsurprisingly, among 46 countries that reported SARS-CoV-2 genomes in January 2021, 40 countries reported a VOC genome from this period. We estimated the ratio of the VOC genomes to all genomes in these countries (Table 1). We found that, in addition to England, in 10 countries the VOC genomes constituted more than half of reported genomes. The data suggest that the strain is spreading globally, even though from countries other than England less than 20% of VOC genomes were reported (as of 16 February 2021).

Using the same method as in the previous section we estimated the replicative advantage of the VOC strain in four other countries in which the number of reported VOC genomes permits such analysis. In Denmark, Scotland, and Wales, similarly to England, the A20.EU1 strain constitutes a large share of all genomes in November 2020, respectively: 39%, 64%, and 70%. We thus compared the VOC with the A20.EU1 strain in these countries. In the USA, where we have not found a dominating D614G substrain, we compare the VOC with all non-VOC genomes. In all four countries, we found periods of the exponential growth of the ratio of compared genomes (seven-week-long in Denmark, Scotland, and Wales, and five-week-long in the USA; see linear growth in logarithmic scale in Figure 3). This allowed us to estimated the replicative advantage of the VOC strain in a narrow range: from $1.69^{6.73/7}$ = 1.65 for Wales to $1.76^{6.73/7}$ = 1.72 for the USA with 95% confidence intervals in the range [1.46–2.04]. We should notice that in all four countries the replicative advantage of the VOC strain has been found smaller than estimated for England.

### 3.4. Replicative Advantage of the 501Y.V2 Strain in South Africa

In addition to Δ69–70HV, mutation N501Y in the RBD of spike is considered as the most important recent mutation [9]. This mutation occurred independently in the South African strain 501Y.V2, where it is accompanied by two other mutations in spike RBD: K417N and E484K [7]. Using the same method as previously we estimated the replicative advantage of the 501Y.V2 strain over other South African strains (Figure 4). In weeks 43–50 of 2020, when an exponential growth is observed, the ratio of 501Y.V2 strain to other strains grows at the weekly rate of 1.58 [95% CI: 1.45–1.72], which gives $R_{t}^{501Y.V2}/R_{t}^{\mathrm{other}}$ = $1.58^{6.73/7}$ = 1.55 [95% CI: 1.43–1.69]. The noisiness of data is associated with the small number of genomes sequenced in the entire period of weeks 43–50 (only 407 501Y.V2 sequences and 376 non-501Y.V2 sequences collected in this period were submitted to GISAID).

### 3.5. Emergence of Mutations in VOC Genomes in England

Because of its high replicative advantage, the VOC strain will likely become globally dominant, possibly reaching fixation. It is thus crucial to track mutations that arise in this strain, that could further increase its replicative advantage. We thus performed sequence analysis of the spike gene in all genomes from England submitted to GISAID till 12 February 2021 (Appendix A, sheet ‘Mutations’). There are 2232 different mutations in genomes collected in England, including 1213 different mutations in VOC genomes with 697 ‘confirmed mutations’ found in more than one submitted VOC genome. Majority of these mutations (535 out of 697) have arisen also in non-VOC strains. This suggest that the mutational space is to a large extent already explored, however the nine mutation characterizing the spike protein of VOC may increase or decrease the replicative advantages of recurrent mutations or allow for propagation of novel mutations. In Figure 5A we show accumulation of mutations is VOC spike in time; within 53,185 analyzed VOC genomes, 20% have at least one mutation, while 1.7% has two mutation or more. In relation to the first collected VOC lineage genome (GISAID accession ID: EPI_ISL_601443), the VOC genomes collected in the end of January have on average 0.3 mutation in their spike protein (Figure 5B).

In Figure 6 we analyze the replicative advantage of 60 VOC substrains for which there were at least 30 genomes submitted to GISAID from England. Each substrain is marked by disk on the (first collection date, total occurrences) plane. The VOC substrains that from time of its first collection grow faster than the VOC strain on average are above the red line that shows the average growth of the VOC strain sequences. Unsurprisingly, the majority of substrains that exceed 30 submitted genomes fall into this category. This analysis should be taken with great caution, as the total number of substrains is high, some of may grow faster in the considered time window just by chance without having any replicative advantage. Nevertheless, one can use this approach to screen for further analysis the substrains that can potentially have a replicative advantage. The most prevalent mutation, defining a fast growing variant is the L18F substitution (1186 genomes) in the N-terminal domain (NTD) of spike protein. The second most prevalent is L5F (658 genomes) localized in the signal peptide of spike. This mutation (at a highly homoplasic position that may be a sequencing artifact [18]) was found abundant also in non-VOC genomes. The third most prevalent mutation, that is also by far the most prevalent RBM as well as in whole RBD mutation, is S494P (441 genomes). The other two fast growing “sibling” mutations in VOC—Q677H (256 genomes) and Q675H (86 genomes)—are present in the the proximity of the polybasic cleavage site (residues 682–685) at the S1/S2 boundary influencing RBD—ACE2 binding [19]. Mutations at residue Q677 (either Q677H or Q677P) were found in several independent lineages spreading over the autumn of 2020 and into the winter of 2021 in the USA [5].

### 3.6. Growth of the L18F Substrain

The first occurrence of the spike L18F substitution has been reported in a VOC strain genome collected on 4 December 2020 (GISAID ID: EPI_ISL_720875). As of 12 February 2021, as much as 1186 spike L18F VOC genomes have been reported in England. Of note, in Autumn 2020, that is, before the VOC lineage has become the dominant strain, the L18F substitution was a ubiquitous mutation in England. Till 12 February 2021, most of the L18F non-VOC genomes in England (97.6%, 25,655 out of 26,280) were found within the 20A.EU1 strain. The fraction of spike L18F mutation in the expanding 20A.EU1 strain was slowly increasing from 35% (1332 out of 3799) in September, 43% (5658 out of 13,046) in October, till 52% (8917 out of 17,470) in November 2020, which may suggest that this mutation was beneficial for the 20A.EU1 strain.

In Figure 7 we show the exponential growth of the L18F VOC substrain in England in the five-week period of 7 December 2020–17 January 2021, in relation to the VOC genomes non-mutated at residue 18, denoted L18. In the considered period, the ratio of the L18F to the L18 genomes increased with the fitted weekly growth rate of 1.75, which gives $R_{t}^{L18F}/R_{t}^{L18}$= $1.75^{6.73/7}$ = 1.72 [95% CrI: 1.57–2.02]. This credible interval is calculated assuming a binomial distribution of the number of the L18F and L18 VOC genomes in each week (see Methods). The confidence interval calculated from 1.96× standard error of the slope, [1.63–1.81], is narrower, which means that the nearly perfect co-linearity of six data points is somewhat coincidental, and the binomial distribution-based credible interval is the proper estimate. This analysis suggests an advantage of the L18F VOC substrain in relation to the remaining VOC genomes, but since it is based on short period, it must be taken with caution. The advantage of L18F substrain is supported by data from Wales, UK, where the L18F VOC genomes constituted 17% (390 out of 2301) of all VOC genomes reported in January, substantially more than in the same period in England, 3.0% (1333 out of 43,700). The number of genomes in Wales is however too small to perform an analysis analogous to that in Figure 7.

### 3.7. VOC Strain Mutations in Spike Receptor-Binding Domain

Of particular concern are the VOC strain mutations occurring in the receptor-binding domain (RBD, residues 333–527), especially mutations in the receptor-binding motif (RBM, residues 438–506). These mutations may potentially lead to immune escape mutants, resulting in reinfection of convalescent individuals and aggravation of the efficacy of current vaccines. Propagation of such mutations is facilitated by high replicative advantage of the VOC strain and potential selection due to the increasing number of convalescent or immunized individuals. The VOC-202012/01 strain spike RBM mutations of special concern are substitutions E484K and S494P.

E484K. A first genome has been collected on 17 December 2020 (GISAID ID: EPI_ISL_782148), and there were 30 genomes reported from the England up till 12 February 2021. The same mutation has occurred in the fast expanding South African and Brazilian (Manaus) strains that share with the VOC substitution N501Y and additionally have a mutation of residue 417: either K417N (South African strain 501Y.V2) [7] or K417T (Manaus strain P.1) [8]. It was suggested that E484K may compromise binding of class 2 neutralizing antibodies, while the A501V mutation interferes with binding of class 1 antibodies. The P.1 strain led to the surge of infections in Manaus in December 2020 despite high seroprevalence of the population (a study of blood donors indicated that 76% [95% CI: 67–98%] of the population in Manaus had been infected with SARS-CoV-2 by October 2020 [20]).

S494P. A first genome has been collected on 12 November 2020 (GISAID ID: EPI_ISL_741039), and there were 441 genomes reported from England up till 12 February 2021. In an in silico study, this substitution has been found to increase complementarity between the RBD and ACE2 [21]. This mutation has been also characterized as an escape mutation by Koenig et al. [22], who also distinguished five additional “escape” residues in the RBM: G447, Y449, L452, F490, G496, and six outside the RBM but within the RBD: Y369, S371, T376, F377, K378, R403. Among these residues, until 12 February 2021, substitution F490S (first collected on 13 December 2020, GISAID accession ID: EPI_ISL_736026) was reported in the highest number of genomes (28 genomes in England).

## 4. Discussion

The mutations of SARS-CoV-2 that substantially increase replicative advantage of emerging strains will likely become dominant, either locally in countries or continents, or worldwide. Substitution D614G in spike protein (first GISAID reported genome, EPI_ISL_913915, was collected on 2 January 2020, in Mexico) initiated a strain with replicative advantage over D614 strains estimated based on data from England as 1.42 [95% CI: 1.38–1.45]. The D614G strain has spread worldwide nearly reaching fixation; it was present in more than 99% of genomes collected worldwide in January 2021. The 20A.EU1 strain, a substrain of D614G that harbors A222V mutation in spike, emerged in Spain in early summer, 2020, spread over Europe, becoming the dominating strain (more than half of sequenced genomes) in several countries (Spain, England, Scotland, Wales, Ireland and Italy) in November, 2020, but was nearly absent outside of Europe. Based on GISAID data in England we estimated its replicative advantage over other non-VOC D614G English strains as 1.24 [95% CI: 1.22–1.27]. The VOC strain started spreading in England in October 2020, outcompeting the A20.EU1 strain, and reached about 80% of genomes in England in January 2021. We have estimated that its replicative advantage over the A20.EU1 strain is in the range 1.83–2.18. The lower bound was obtained by fit in the eight-week-long period of weeks 43–51 of 2020, when the ratio of the VOC to the A20.EU1 strain genomes increased 217 times, from 0.0074 to 1.62, whereas the upper bound was obtained in the four-week-long period of weeks 43–47 of 2020. We think that the slower growth in the period of weeks 47–51 is a consequence of (1) the fact that in Kent, Greater London, and their vicinity, the VOC strain almost reached fixation and (2) the fact that in these regions more stringent measures were implemented to suppress rapid growth of cases.

We also estimated the replicative advantage of the VOC strain in relation to the 20A.EU1 strain in Denmark, Scotland, and Wales, and in relation to bulk non-VOC strains in USA. We found find values in range from 1.69 for Wales to 1.76 for USA, with 95% confidence intervals in range [1.46–2.04], that are smaller than in England. One possible explanation is that the VOC strain is able to infect seroprevalent or exposed individuals. Such ability would increase its replicative advantage in a population, in which the fraction of seroplevalent individuals is large [23]. This mechanism would also help to explain the higher replicative advantage of the VOC strain observed in weeks 43–47 of 2020, when the VOC strain was gaining prevalence in London area [3]. The strain P.1, found in Brazil, that shares with VOC RBM mutation N501Y, caused recently the second (higher) wave of deaths in Manaus despite high seroprevalence of the population [8].

In addition to double deletion Δ69–70HV, substitution N501Y in the RBD of spike is considered the most important VOC mutation [9]. This mutation occurred independently in the South African strain 501Y.V2. We estimated that 501Y.V2 has replicative advantage over other South African strains equal 1.55 [95% CI: 1.43–1.69]. The replicative advantage of 501Y.V2 strains supports the conjecture that mutation N501Y increases infectiousness of SARS-CoV-2 by increasing the affinity of spike RBD to the angiotensin-converting enzyme 2 (ACE2) [24].

Both our estimates suggests that the replicative advantage of the VOC strain is higher than early estimate, 1.47 [95% CI: 1.34–1.59] [9]. Also Davies et al. [11] estimated that the VOC strain has 43–82% [95% CrI: 38–106%] higher transmissibility. In these studies, the authors estimated the replicative advantage of the VOC strain separately for each region and then averaged over analyzed regions. We think that our approach, based on weekly and England-averaged data, gives a more accurate estimate. This is because the VOC strain (as well as other strains) evolves and some substrains have a lower or even no replicative advantage and will become extinct in the course of evolution, and finally a substrain with the highest replicative advantage will dominate. Substrains with a small replicative advantage may contribute to the VOC to non-VOC replication ratio averaged over NHS STP areas of England but only marginally influence the expansion of the VOC strain globally. By using the aggregated approach we estimate the replicative advantage of the dominating substrain(s). To convert the weekly growth of the ratio of genomes to the ratio of respective reproduction numbers we assumed that both strains have the same mean serial interval equal 6.73 days. This calculation is approximate as the serial interval is not a number but follows a hypoexponential distribution [17]. Additionally, although no current data indicate this, it may happen that the faster spread of VOC strain partially results from a shorter generation time.

The VOC strain, because of its high replicative advantage, is likely to become globally dominant, possibly reaching fixation. It will continue to evolve so it is crucial to track which mutations already present in this strain can further increase its replicative advantage. We thus performed detailed sequence analysis of the spike protein identifying, as of 12 February 2021, 1213 different mutations in VOC genomes collected in England with 697 of them found in more than one submitted VOC genome. By systematic analysis of the propagation of VOC substrains we found that substrain(s) conferring L18F substitution is/are the most abundant and rapidly growing VOC substrain(s).

The data collected in the five-week period of 7 December 2020–17 January 2021, in England suggest a replicative advantage of the L18F substrain in relation to the remaining VOC strains. The estimate of the replicative advantage of 1.72 [95% CrI:1.57–2.02] is based on a relatively short time interval so it must be taken with caution. Importantly, the L18F mutation has also expanded in the South African strain 501Y.V2 defined by three spike mutations K417N, E484K, N501Y (thus sharing with the VOC strain spike mutation N501Y). Among the 501Y.V2 genomes collected after 1 December 2020, the L18F substrain constitutes 41% genomes (127 out of 309), according to GISAID as of 12 February 2021). In Brazil, in strain P.1 defined by three spike mutations K417T, E484K, N501Y (differing from the South African strain 501Y.V2 by substitution K417T instead of K417N), mutation L18F has been found in 93% of genomes (69 out of 74) collected after 1 December 2020. This data suggests a replicative advantage of L18F substrains within the VOC, 501Y.V2, and P.1 strains, in, respectively, England, South Africa, and Brazil. This replicative advantage of the VOC L18F substrain must be considered with caution until the mechanism promoting faster spread of strains containing L18F substitution is elucidated. Leucine 18 lies in the N-terminal domain (NTD), that has not been typically considered as a target for neutralizing antibodies. However, there is a growing number of studies showing that the NTD is targeted by antibodies and that NTD deletion 69H–70V (characterizing the VOC strain) compromises binding of antibodies [25,26,27]. With respect to L18F, an in vitro study by Cele et al. shows that an African variant L18F, D80A, D215G, K417N, E484K, N501Y, D614G, A701V propagates much faster than a variant without L18F mutation in the presence of plasma antibodies collected from donors infected in the first wave of epidemic in South Africa (June–August 2020) [28]. Correspondingly, McCallum et al. showed that L18F substitution compromises binding of neutralizing antibodies [15]. Findings by Cele et al. and McCallum et al., together with the increase of L18F variants in 501Y.V2, P.1, and VOC strains, suggests that the replicative advantage of L18F mutants can be partly associated with their ability to infect seroprevalent individuals, and thus depend on the fraction of seroprevalent individuals in given territory. In turn, growth of strains with mutations in escape residues L18 and S494 on the VOC strain suggests an increasing selection pressure resulting from the growth of the seroprevalent fraction of the population of England. This trend can be enhanced by the ongoing English vaccination program, in which the relatively large time span between the first and second dose can be a contributing factor.

In summary, we have shown that the new VOC strain has about twofold replicative advantage over the 20A.EU1 strain of SARS-CoV-2, that was dominating in England in November 2020. The strain has already spread across the world and will likely spread further towards fixation. It was present in 40 out of 46 countries that reported at least 50 viral genomes in January, 2021. Spread of the faster-replicating VOC-202012/01 strain may hinder the efforts to contain the COVID-19 epidemics prior to mass vaccinations. As the global spread of the VOC strain is very likely, it is important to monitor mutations of this strain, with particular attention to mutations interfering with immune response including the fast spreading NTD mutation L18F, and RBM mutations E484K, F490S, and S494P that may decrease the efficacy of currently available vaccines.

## Figures and Tables

**Figure 1 viruses-13-00392-f001:**
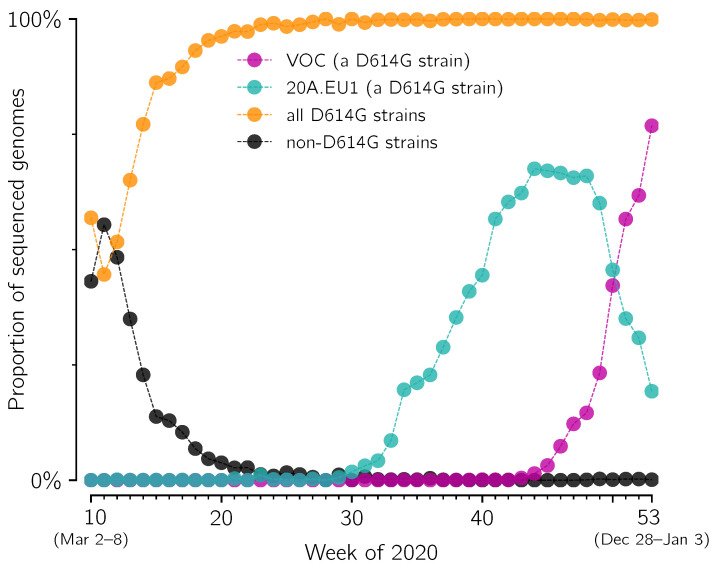
Fixation of D614G strain and competition between its substrains: 20A.EU1 and the VOC in England. The proportion of D614G (orange), 20A.EU1 (cyan), VOC (magenta) and D614 strains (black) in weeks 10–53 of 2020 has been determined based on GISAID data available on 12 February 2021 (provided in Appendix A).

**Figure 2 viruses-13-00392-f002:**
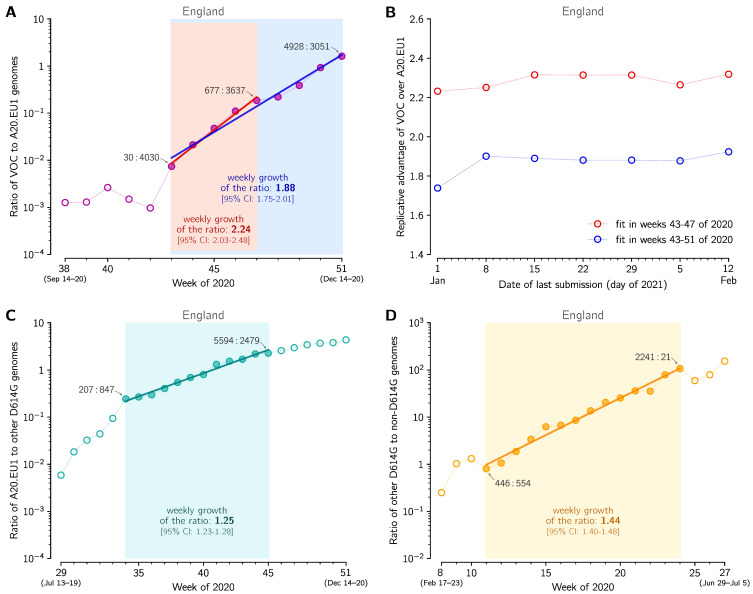
The replicative advantage of the VOC, 20A.EU1, and the D614G strain. (**A**) The ratio of the VOC to 20A.EU1 sequences collected in weeks 38–51 of 2020 in England. The trend line is fitted to data points from weeks 43–51 (blue) and from weeks 43–47 (red). The weekly growth rate is 1.88 [95% CI: 1.75–2.01] for weeks 43–51 and 2.24 [95% CI: 2.03–2.48] for weeks 43–47. (**B**) Stabilization of fits shown in panel A with time-shifting date of the last submission. (**C**) The ratio of the 20A.EU1 to non-20A.EU1 non-VOC D614G genomes collected in weeks 29–51 of 2020 in England. The trend line is fitted to data points from weeks 34–45. The weekly growth rate is 1.25 [95% CI: 1.23–1.28]. (**D**) The ratio of the D614G to D614 genomes collected in weeks 11–27 of 2020 in England. The trend line is fitted to data points from weeks 11–24. The weekly growth rate is 1.44 [95% CI: 1.40–1.48]. Data points for selected weeks are labeled with ratios of the counts of genome sequences. Panels A, C and D are based on GISAID data submitted till 12 February 2021 (provided in Appendix A).

**Figure 3 viruses-13-00392-f003:**
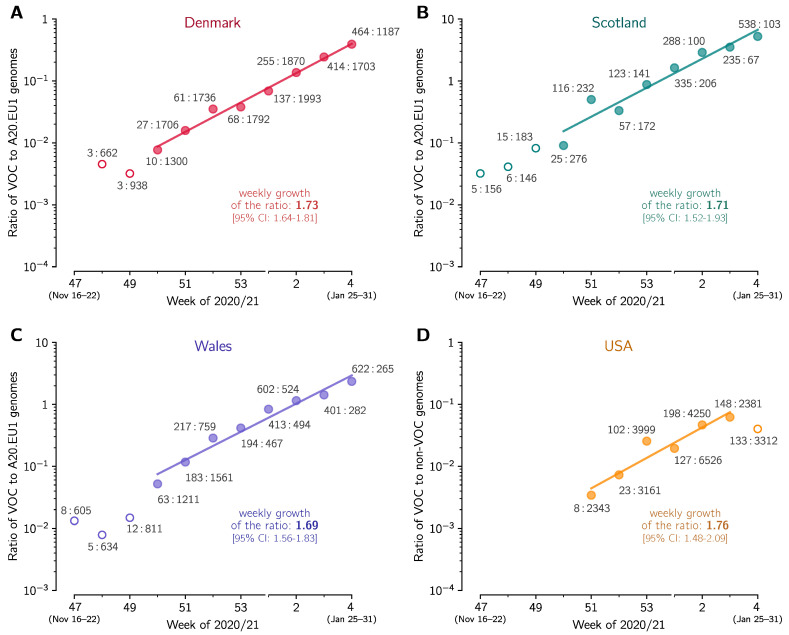
The replicative advantage of the VOC in (A) Denmark, (B) Scotland, (C) Wales, and (D) USA. (**A**) The ratio of the VOC to the 20A.EU1 sequences collected in Denmark. The trend line is fitted to data points from week 50 of 2020 to week 4 of 2021. (**B**) The ratio of the VOC to 20A.EU1 sequences collected in Scotland. The trend line is fitted to data points from weeks 50 of 2020 to week 4 of 2021. (**C**) The ratio of the VOC to 20A.EU1 sequences collected in Wales. The trend line is fitted to data points from week 50 of 2020 to week 4 of 2021. (**D**) The ratio of the VOC to non-VOC sequences collected in USA. The trend line is fitted to data points from week 51 of 2020 to week 3 of 2021. Data points are labeled with ratios of the counts of genome sequences. The weekly growth rates and respective confidence intervals are given in each panel. All panels are based on GISAID data submitted till 17 February 2021 (provided in Appendix A).

**Figure 4 viruses-13-00392-f004:**
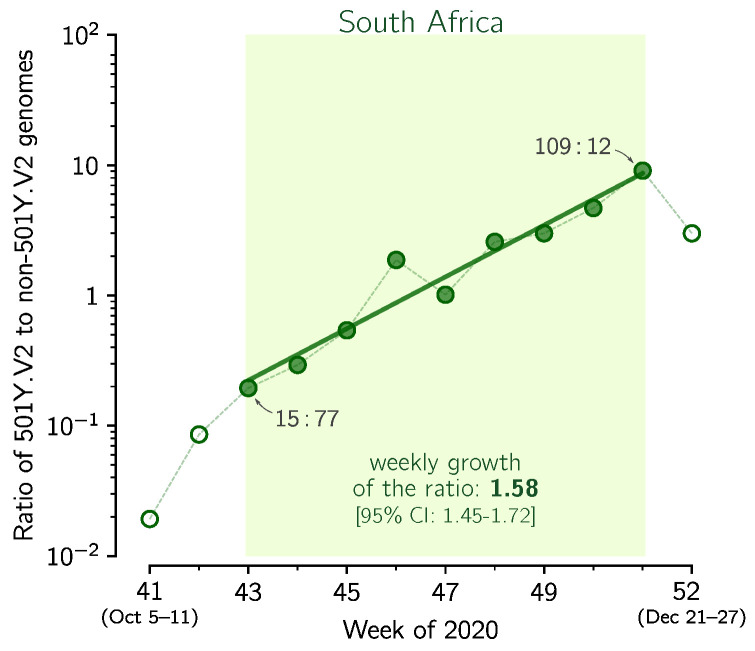
The replicative advantage of the 501Y.V2 strain in South Africa. Shown is the ratio of the 501Y.V2 to the non-501Y.V2 genomes collected in weeks 41–50 of 2020 in South Africa. The trend line is fitted to data points from weeks 43–50. The weekly growth rate is 1.58 [95% CI: 1.45–1.72]. Data points for selected weeks are labeled with ratios of the counts of genome sequences. The figure is based on GISAID data submitted till 12 February 2021 (provided in Appendix A).

**Figure 5 viruses-13-00392-f005:**
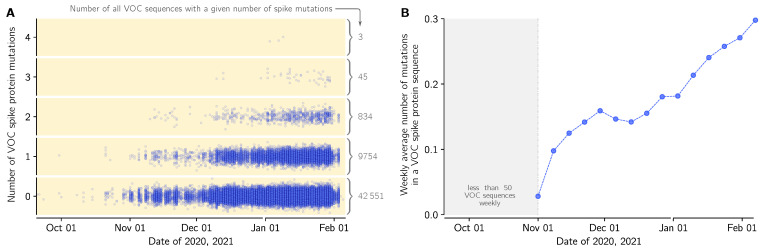
Mutations in spike protein in the VOC lineage in England in relation to the first VOC genome collected in September 20, 2020 (GISAID sequence accession ID: EPI_ISL_601443): (**A**) Dots denote genomes collected in a given date (horizontal axis) with a respective number of novel (amino acid-level) mutations in relation to the first VOC genome (vertical axis). The number of the genomes with a given number of novel mutations is provided as gray numbers next to a brace. (**B**) The average number of novel (amino-acid level) mutations in a VOC lineage sequence (weekly average).

**Figure 6 viruses-13-00392-f006:**
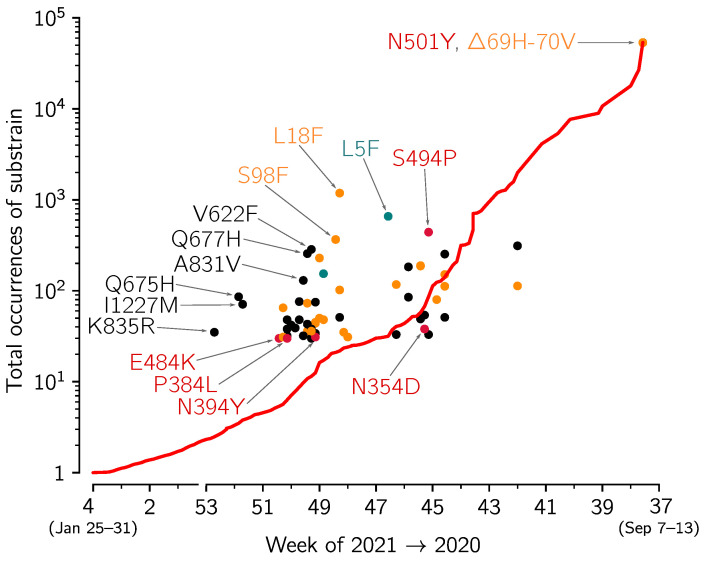
The growth of VOC substrains with respect the whole VOC strain. The figure is based on spike protein sequences from genomes collected in England and submitted to GISAID till 12 February 2021. The growth of the whole VOC strain (red line) in the time span between *t* and $t_{0}$ = 12 February 2021 is calculated as VOC($t_{0}$)/VOC(*t*), i.e., the ratio of the number of the VOC genomes collected till 12 February 2021 and the number of VOC genomes collected till *t* (date given on horizontal axis). Disks denote VOC substrains defined by particular mutations (with at least 30 genomes submitted); horizontal axis value for each circle is the date of first collection of a sequence with a given mutation; vertical axis value is the number of genomes collected till 12 February 2021, conferring a given mutation. Red, orange, and green circles denote substrains with leading mutation in respectively in RBD, NTD, and signal peptide; black circles denote mutations in other domains. Selected substrains that may have replicative advantage and all substrains with the leading mutation in RBD are annotated. Detailed data that were analyzed to prepare this figure are provided in Appendix A (sheet ‘Mutations’).

**Figure 7 viruses-13-00392-f007:**
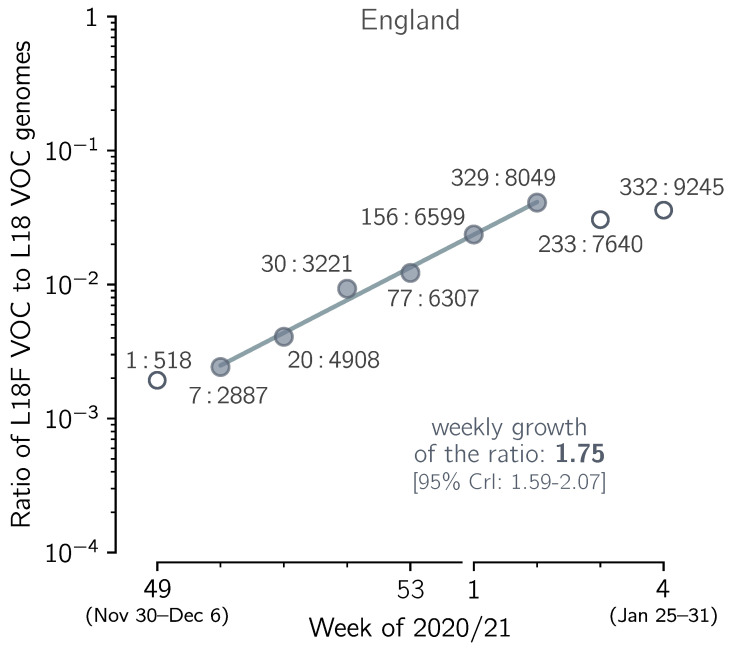
The growth of the spike L18F substrain in relation to the L18 VOC strains. The ratio $v_{L}$ of the number of VOC genomes conferring spike L18F mutation to the number of non-mutated (L18) VOC genomes collected in the period between week 49 of 2020 and week 3 of 2021 in England. Data aggregated into weeks indicated that $v_{L}$ changes from 7:2887 = 0.0024 in week 50 of 2020 to 329:8049 = 0.041 in the second week of 2021, i.e., 16.9 times, meaning that $v_{L}$ increases $16.9^{1/5}$ = 1.76-fold per week. The trend line is fitted to data points shown as filled circles. The weekly growth rate of the ratio is 1.75. The 95% CI calculated as 1.96 × standard error of the slope is [1.66–1.85]; the 95% CrI calculated assuming binomial distribution of substrain genomes is [1.59–2.07]. Data points in week 50 of 2020 and week 2 of 2021 are labeled with ratios of the counts of both types of genome sequences. The figure is based on GISAID data submitted until 12 February 2021 (provided in Appendix A).

**Table 1 viruses-13-00392-t001:** Fraction of VOC genomes in countries that in January 2021 reported more than 50 genomes.

Country	Proportion of VOC Genomes		95% CrI
England	86.1%	(43,260/50,240)	85.8–86.4%
Scotland	71.3%	(1445/2028)	69.3–73.2%
Ghana	68.9%	(62/90)	58.9–77.8%
Turkey	68.6%	(155/226)	62.4–74.3%
Slovakia	67.7%	(42/62)	56.5–79.0%
Ireland	62.0%	(364/587)	58.1–65.9%
Italy	61.0%	(169/277)	55.2–66.8%
Nigeria	60.3%	(70/116)	51.7–69.0%
N. Ireland	57.9%	(183/316)	52.5–63.3%
Wales	53.5%	(2125/3971)	52.0–55.1%
Romania	52.0%	(26/50)	38.0–66.0%
Spain	49.1%	(576/1174)	46.2–52.0%
France	44.7%	(691/1545)	42.3–47.2%
Israel	36.9%	(113/306)	31.7–42.5%
Australia	34.4%	(66/192)	27.6–41.2%
Austria	34.2%	(178/520)	30.2–38.3%
Belgium	32.6%	(538/1651)	30.4–34.9%
Czechia	32.1%	(17/53)	20.8–45.3%
Finland	31.8%	(67/211)	25.6–37.9%
Sweden	27.7%	(109/394)	23.4–32.2%
Netherlands	21.0%	(328/1561)	19.0–23.1%
Singapore	20.6%	(30/146)	14.4–27.4%
Portugal	20.0%	(127/636)	17.0–23.1%
Germany	16.9%	(68/403)	13.4–20.6%
Sri Lanka	14.8%	(13/88)	8.0–22.7%
Norway	14.3%	(27/189)	9.5–19.6%
S. Korea	13.3%	(12/90)	6.7–21.1%
India	12.9%	(11/85)	5.9–20.0%
Luxembourg	10.7%	(27/253)	7.1–14.6%
Denmark	10.5%	(1270/12,137)	9.9–11.0%
Brazil	9.5%	(13/137)	5.1–14.6%
N. Macedonia	9.3%	(5/54)	1.9–18.5%
Switzerland	8.2%	(409/4985)	7.4–9.0%
Iceland	7.7%	(5/65)	1.5–15.4%
Canada	7.0%	(36/514)	4.9–9.3%
Poland	6.7%	(11/165)	3.0–10.9%
Mexico	6.5%	(5/77)	1.3–13.0%
Latvia	5.7%	(5/87)	1.2–11.5%
USA	3.7%	(608/16,439)	3.4–4.0%
S. Africa	0.9%	(1/112)	0.0–2.7%
Colombia	0.00%	(0/64)	
Mayotte	0.00%	(0/224)	
Lithuania	0.00%	(0/381)	
Egypt	0.00%	(0/84)	
Kenya	0.00%	(0/75)	
French Guiana	0.00%	(0/98)	

## Data Availability

Publicly available datasets were analyzed in this study. This data can be found at https://www.gisaid.org (available after registration; accessed on 12 and 17 February 2021, see Methods) and https://www.cogconsortium.uk (accessed on 15 February 2021), and at https://www.ecdc.europa.eu/en/publications-data/data-national-14-day-notification-rate-covid-19 and https://coronavirus.data.gov.uk/details/cases (accessed on 17 February 2021). We also provide retrieved and preprocessed data in a single Appendix A.

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
