# Peer review of "SARS-CoV-2 Variant of Concern 202012/01 Has about Twofold Replicative Advantage and Acquires Concerning Mutations"

_viruses, 2021, doi:10.3390/v13030392_

Round 1

Reviewer 1 Report

The article „SARS-CoV-2 Variant of Concern 202012/01 has about twofold replicative advantage” investigates the replicative advantage of different SARS-Cov-2 variants with a focus on the UK mutant, which started spreading rapidly in mid-October 2020. The conclusions are supported by the calculation results based on the GISAID database. The results have some relevance, but I am not entirely sure that the article is powerful enough to publish it in a journal with an IF ~4. Regardless, there are several points where the article could be improved.

The writing of the article feels rushed and proofreading by a native English speaker is recommended. Furthermore, the structuring of the article could be improved, in my opinion there are parts in the introduction and results that belong to the discussion.

The sentence, which starts at line 50 should have some reference or a little more explanation on how SGTF can be used as a proxy for the prevalence of the VOC genome.

In line 119-120 it is not stated that the 1.85 [95% CI: 1.74–1.97] ratio is calculated for how many weeks.

The description of the methods is lacking. The authors should provide the names of software and packages (e. g. in case of R) applied for modelling. The code used for calculations could also be provided in the supplementary material, so others could repeat and validate the calculations.

Tables should have a title and a caption.

In line 252-253, I think that instead of 501Y.V2, there should be 20A.EU1 or it does not really make sense.

How does the number of diagnostic tests performed in the UK or South Africa may influence these results?

How many sequences were uploaded to the GISAID database compared to the confirmed COVID cases, and how its variability may influence the results?

Davies et al. (10.1101/2020.12.24.20248822) already made a similar study, but they used a different database (https://www.cogconsortium.uk/data/). Is there a major difference between this database and GISAID, which could have an impact on the results?

Reviewer 2 Report

The manuscript by Grabowski et al., was a relatively comprehensive research article on the potential effects of genomics variation on the population with SARS-CoV-2 infection. The authors focused on the several mutation strains in the SARS-CoV-2 contaminated samples. Investigations were well performed and data was well collected. There were some moderate concerns:

  • It is quite understood that the viruses have been regularly monitored. However, the rationale of studying “period between week 43 and week 51 of 2020” was unclear.
  • There was no statistical analysis on the Figure 1 and Table 1.
  • Lines 169-170, “countries will likely experience a surge of COVID-19 cases” seemed not to belong to the Results section.
  • Please add discussions on what data can mean specifically regarding the source of data.

Round 2

Reviewer 1 Report

The article is significantly improved after the revision. I do not have more concerns.